# Design and Evaluation of a Flexible Dual-Band Meander Line Monopole Antenna for On- and Off-Body Healthcare Applications

**DOI:** 10.3390/mi12050475

**Published:** 2021-04-22

**Authors:** Shahid M Ali, Cheab Sovuthy, Sima Noghanian, Zulfiqur Ali, Qammer H. Abbasi, Muhammad A. Imran, Tale Saeidi, Soeung Socheatra

**Affiliations:** 1Department of Electrical and Electronic Engineering, Universiti Teknologi, PETRONAS Bander Seri Iskandar, Tronoh 32610, Perak, Malaysia; shahid_17006402@utp.edu.my (S.M.A.); tale_g03470@utp.edu.my (T.S.); socheatra.s@utp.edu.my (S.S.); 2Wafer LLC, 2 Dunham Rd, Beverly, MA 01915, USA; 3American Public University System, 111 W Congress St, Charles Town, WV 25414, USA; 4Healthcare Innovation Centre, School of Health and Life Sciences, Teesside University, Middlesbrough, Tees Valley TS1 3BX, UK; Z.Ali@tees.ac.uk; 5James Watt School of Engineering, University of Glasgow, Glasgow G12 8QQ, UK; qammer.abbasi@glasgow.ac.uk (Q.H.A.); muhammad.imran@glasgow.ac.uk (M.A.I.); 6Artificial Intelligence Research Center (AIRC), College of Engineering and Information Technology, Ajman University, Ajman, United Arab Emirates

**Keywords:** WBAN, wearable antenna, on- and off-body communications, SAR

## Abstract

The human body is an extremely challenging environment for wearable antennas due to the complex antenna-body coupling effects. In this article, a compact flexible dual-band planar meander line monopole antenna (MMA) with a truncated ground plane made of multiple layers of standard off-the-shelf materials is evaluated to validate its performance when worn by different subjects to help the designers who are shaping future complex on-/off-body wireless devices. The antenna was fabricated, and the measured results agreed well with those from the simulations. As a reference, in free-space, the antenna provided omnidirectional radiation patterns (ORP), with a wide impedance bandwidth of 1282.4 (450.5) MHz with a maximum gain of 3.03 dBi (4.85 dBi) in the lower (upper) bands. The impedance bandwidth could reach up to 688.9 MHz (500.9 MHz) and 1261.7 MHz (524.2 MHz) with the gain of 3.80 dBi (4.67 dBi) and 3.00 dBi (4.55 dBi), respectively, on the human chest and arm. The stability in results shows that this flexible antenna is sufficiently robust against the variations introduced by the human body. A maximum measured shift of 0.5 and 100 MHz in the wide impedance matching and resonance frequency was observed in both bands, respectively, while an optimal gap between the antenna and human body was maintained. This stability of the working frequency provides robustness against various conditions including bending, movement, and relatively large fabrication tolerances.

## 1. Introduction

Currently, advances in the miniaturization of wireless devices and designs of smart wireless networks have allowed a rapid development in wireless body area networks (WBAN) because of their vast demands for numerous applications such as those in healthcare, the military, sports, and electronic gaming [1]. These applications require an easy integration of flexible and textile-based high-data-rate wireless electronic devices into clothing and other wearable gadgets, so that the wearer can easily communicate with various devices [2]. One of the key components for body-centric wireless communication (BCWC) is the antenna. BCWC has to work near the human body; therefore, the wearable antenna must meet certain requirements such as being planar, compact, and flexible, and it should be able to easily be integrated with electronic systems and maintain a reliable link while not causing any discomfort for the user. The human body is a hostile environment for an antenna due to the coupling and absorption of the EM waves [3]. Wearable antennas must be designed and characterized carefully in order to maintain a reliable communication link even under the detuning effects of lossy body tissues, because the human body can deteriorate the performance of the antenna [4]. To add to this complexity, it is worth noting that every individual has a composition of body tissue with different dielectric constants (ε) and geometrical parameters [5]. However, the effect of coupling in the human body and wearable antennas can be mitigated by using de-coupling methods such as insolation and proper use of the ground layers. Small antennas with a truncated or no ground plane generally are omnidirectional, and therefore, are more affected by the body’s coupling effect. This effect requires optimizing the antenna-body separation distance effectively [6]. The study and optimization of the antenna with small isolation from the body provide a simple and effective way of combating the detuning effect. Figure 1 shows the general architecture in WBAN systems.

Numerous flexible textile antennas have been designed and evaluated in WBAN systems. For example, Ala Alemaryeen et al. integrated a planar monopole antenna with a layer of the artificial magnetic conductor (AMC) in [8]. The antenna was completely made of fabric using Pellon for the substrate, and pure copper taffeta for conducting parts. An impedance bandwidth of 4.30–5.90 GHz was observed. Effects of bending and crumpling were studied and the performance of the design on various body models was compared. Stephen J. Boyes et al. in [9] designed an inverted-F antenna on a felt substrate, and its performance was checked on various body locations. The impact of different materials, as well as bending, was studied. The bending effects significantly degraded the antenna’s performance by changes in its distance from the body. Dominique L. Paul et al. in [10] investigated a wideband antenna, working within the band of 0.9–6.0 GHz. The antenna performance was investigated under bending conditions, which showed a considerable change in the antenna’s resonant frequency. In addition, the antenna’s efficiency was improved by adding an electromagnetic bandgap (EBG) surface.

In our work, a compact flexible dual-band meander line monopole antenna is proposed, and its performance was evaluated on different subjects with various bending and wet conditions, and at two operating bands with different body sizes, in which a wearable antenna is expected to operate for on-/off-body applications. Table 1 shows the comparison of this design with similar results from a few papers from the literature and improvements that were achieved. In particular, we have shown that the results are not sensitive to the presence of the human body, and insignificant detuning is observed at both bands when a 10 mm distance was maintained. Moreover, these evaluations are influential in determining proper antenna operation on the human body, and also pave the way for unique designs as mentioned by Guy A. E. Vandenbosch et al. in [11,12], and Andrea Ruaro et al. in [13] for future high-performance on-body applications. This paper is arranged as follows: after the introduction, Section 2 describes the design steps and the different parameters of the wearable monopole-shaped meander line antenna. The subsections describe the antenna geometry, design strategy, radiation modes, and optimization of the antenna in the on-body model. In Section 3, antenna measurement results are presented. The subsection includes the results of reflection coefficients, bandwidth, far-field radiation, and specific absorption rate (SAR) simulation results. Section 4 describes the equipment setup and procedure, and free-space and on- and off-body transmission losses and link budget analysis. Finally, Section 5 contains the conclusions and future direction.

## 2. Antenna Design

This section describes the antenna design and its radiation modes. The antenna’s design guidelines are given considering the variations in its surface current density and the electric field. The antenna optimization is also considered to create a realistic design for on-body applications.

### 2.1. Proposed Topology and Approach

We started with a meander line planar monopole antenna, as shown in Figure 2. CST Microwave Studio (CST MWS) simulation software was used for all the simulations reported in this paper. The antenna consists of a planar monopole loaded with asymmetrical inverted slots and a truncated ground plane, which are made of ShieldIt™ and copper tape with a thickness of 0.17 mm and 0.035 mm, respectively, and it uses a felt substrate. The felt substrate has a relative permittivity (ε_r_) of 1.3, loss tangent (tan δ) of 0.044, and a thickness of 1.5 mm. A denim layer of ε_r_ = 1.43, tan δ = 0.02, and a thickness of 0.7 mm was used under the ground plane to provide isolation between the antenna and the skin surface. The antenna was excited by a microstrip line connected to a monopole length (L_mon_) of 40.20 mm. By adding the slots and creating a proper loading the antenna’s Q can be lowered, and therefore, the size can be miniaturized. The correct choice of slot dimensions can keep the bandwidth (BW) wide. In this design, the meander line antenna dimensions were calculated using transmission line equations [24,25]. Table 2 shows the important dimensions of the proposed antenna.

### 2.2. Proposed Design Strategy

The design goals were as follows: the first goal was to design a compact dual-band monopole meander line antenna with a flexible truncated ground, incorporating textile materials. The second goal was to show reliable on-/off-body communication with wide radiation patterns in both frequency bands and small SAR values. The design of this antenna started with a meander line attached to a microstrip line on a truncated ground plane and separated by a felt substrate. The antenna yielded a wide impedance BW. The antenna size was reduced by asymmetrical inverted slots loading. The meander line was fabricated using copper tape with a thickness of 0.035 mm and resistance of 0.04 Ω/sq. The ground was made of e-textile (ShieldIt™) with a thickness of 0.17 mm, a weight density of 230 g/m^2^, and a resistivity of 0.05 Ω/sq. Moreover, in our CST MWS simulations, lossy material with a conductivity (σ) of 1.18 × 10^5^ S/m was considered to model the ShieldIt material. Table 3 illustrates the important parameters of the proposed design.

### 2.3. Radiation Modes

The antenna gives two types of radiations in the lower and upper bands. The lower band is used for on-body communication, whereas the upper band is used for off-body communication. In the lower band, it acts as a monopole antenna with vertical omnidirectional radiation, which has less radiation toward the body. The lower band reduces the shadow fading issue because of its strong diffraction and longer wavelength. However, in the upper band, a higher mode of the meander line is combined with the first mode of the monopole strip to obtain a wide bandwidth and omnidirectional radiation pattern. When the antenna is mounted on-body, the radiation pattern becomes wider with a wide bandwidth due to the lossy tissues. Thus, the power is radiated nearly in all directions and provides an omnidirectional radiation pattern in both bands. Figure 3a shows the current path at the lower band, which follows the path of A–B–C–D ($l_{1}$). It demonstrates that the antenna’s length at the resonant frequency is half a wavelength. Figure 3b shows the quarter-wavelength resonance mode, where the current follows around the path of A–B–C–E ($l_{2}$). It can be observed that there is a small current on the ground plane. Thus, the resonant frequencies depend on the ground plane and the long ($l_{1}$) and short ($l_{2}$) parts of the meander line monopole antenna.

The simulated reflection coefficients (S_11_) of various antennas and their effects of length on the overall reflection coefficient are shown in Figure 4. The length of the top monopole-shaped meander line strip and a truncated ground plane has a great impact on the resonant frequency and impedance bandwidth (BW). Increasing or decreasing the length of the monopole strip ($l_{1}$) results in a lower resonant frequency. The length of ($l_{1}$) and a ground plane has a great impact on the operating frequency and impedance BW in the lower band, varying the length from 40.20 mm to 36.49 mm shifted the resonant frequency from approximately 2.8 GHz to 3.4 GHz, while there was a minor effect on the upper impedance bandwidth (BW), which shifted from 5.90 to 6.295 GHz. The length of the meander line strip ($l_{2}$) also affects the resonant frequency and impedance bandwidth (BW) in the upper band. Varying the length from 8.49 mm to 4.49 mm drastically shifted the resonant frequency from approximately 5.7 GHz to 9.0 GHz, pushing it out of the required band for Wi-Fi/WLAN, with a huge effect on the lower band. Moreover, we have examined the antenna structure, in which the monopole strip current is concentrated on the meander line, which further reduces the impact from the bending and provided stability in the design. Therefore, in our proposed antenna, we observed that the S_11_ changes with length, as well as how it shifted the frequency and optimized the dimensions to minimize this shift. The optimized design parameters are listed as above in Table 2.

### 2.4. Optimization for On-Body Communication

To utilize a wearable antenna for on-body communication, it is important to optimize the antenna to work while positioned on the body and lossy tissues, such as the chest or arm. In this section, the antenna’s performance on human chest and arm models will be presented. A 3D-layered body model in CST MWS, containing skin, fat, and muscle layers, was utilized for the chest as a flat body model, while for an arm a layered cylindrical model was used to examine the antenna’s performance. Taking the simulation time and accuracy into consideration, a 3D-layered flat body model with the dimensions of 200 × 200 × 50 mm^3^ and thickness (skin = 4 mm, fat = 8 mm, and muscle = 40 mm) was used, with muscle’s dielectric constant of 52.7 and 48.2, and conductivity of 1.95 and 6 S/m at the lower and upper frequency bands, respectively. For the cylindrical model (human arm), a radius of 50 mm and length of 150 mm were chosen, following the on-body procedure as mentioned by Guy A. E. Vandenbosch et al. in [26,27]. In free-space communication, the antenna showed a good impedance matching in the lower and upper bands. Furthermore, the air-gap between the antenna and body was adjusted at 3, 5, and 10 mm. This choice of air-gap is to mimic utilizing comparable thicknesses of multiple textile layers, which were modeled as an air-gap because textile materials usually have a permittivity close to air. As the antenna was adjusted on a body chest model and the arm of a cylindrical model with an air-gap of d = 3 mm, impedance matching was considerably affected due to the strong mutual coupling. However, as the air-gap was increased, the impedance matching was not much affected. The antenna was nearly decoupled from the body at d = 10 mm; thus, the air-gap was adjusted at this distance. Figure 5 illustrates the simulated reflection coefficient (S_11_) of the meander line antenna at different distances from the body model. The optimized dimensions are highlighted as above in Table 2.

## 3. Measurement Results

Figure 2d illustrates the fabricated antenna. The S_11_ was measured at the Universiti Teknologi Petronas (UTP) antenna measurement facility using a network analyzer (Keysight PNA-X N5242A), which was calibrated using the calibration module (NA4691B) from Keysight. For on-body measurement, a flat human tissue model was used with the same dimensions that were used in the simulation. A gap similar to the assumed air-gap in the simulations of the distance (d) of 10 mm was created between the antenna and body surface. This was achieved using multiple layers of denim textiles to create a layer similar to the thickness of clothes and the air-gap between clothes and the body. This air-gap is also necessary for safety, which is briefly discussed by Zheyu Wang et al. in [28]. Furthermore, to study the effect of the body on the antenna characteristics, the fabricated antenna was tested on various locations on-body. It can be noticed that the body affects the impedance BW and radiation characteristics in two bands, while the antenna showed good matching within the desired frequency range. Table 3 illustrates the performance of the proposed antenna.

### 3.1. Reflection Coefficient (S_11_) and Bandwidth

Figure 6 shows the measured S_11_ for the antenna in three cases: the free-space, placed on the chest, and on the arm. Measured and simulated results agree well. The antenna showed good matching at both bands, and S_11_ in both bands was lower than −10 dB. The antenna was measured while it was placed on two subjects with different weights (65 kg and 110 kg) and showed performance similar to the free-space at both bands. Initially, the antenna was relatively flat on the body. The flatness and air and textile layers between the body and antenna provided similar conditions to a free-space scenario. Therefore, the body size had a modest effect on the antenna’s performance. The additional denim layer under the ground created better isolation from the body; as a result, there was less effect of back radiation, which is radiation toward the body tissues. The air-gap was chosen to be 10 mm. Similarly, the antenna was placed around the arm (with a radius of 50 mm and 100 mm) maintaining the same air-gap; the impedance bandwidth was not much affected due to the reduced coupling effects between them. Two other conditions that might occur are the bending of the antenna and body humidity (sweating) causing moisture to penetrate the antenna. In the cases, when the antenna was placed on the chest or arm, the antenna still covered the WLAN band without any considerable changes in the levels of S_11_. The performance of the antenna under bending conditions was experimentally studied in different locations. The antenna was measured in free-space and on polystyrene cylinders with a dielectric constant of ε_r_
≈ 1, for cylinders with various diameters (D in mm) of 100, 80, 70, 60, and 50. Different diameters gave unique antenna curvatures to test whether the desired operating frequency range was maintained under bending conditions. For all cases, there was a minor shift in the center frequency. Overall, the measurement results show that the antenna works within the required band without any considerable frequency detuning. The measured S_11_ values at different bends are shown in Figure 7.

Wearable devices are often affected by the humidity of the body. We analyzed the case where the proposed antenna was dampened by moisture by applying water to the textile substrate until it was saturated, as shown in Figure 8. The resulting S_11_ was severely affected by the wet condition. After one-hour, further changes in the results occurred due to the evaporation of water from the whole substrate. After 2 h, the measured S_11_ results were similar to those measured after one hour, which showed that the evaporation was completed after one hour. As more time passed and the antenna slowly dried and cooled back to the room temperature of 26 °C along with an air humidity of 30%, the measured S_11_ values were similar to those measured before adding the moisture.

### 3.2. Radiation Patterns

The far-fields were measured in the anechoic chamber at the UTP’s antenna measurement facility, as shown in Figure 9. The meander line antenna was placed on a rotating stand and worked as a receiving antenna, whereas the horn antenna moved in the azimuth and elevation planes and worked as a transmitting antenna. Thus, the radiation patterns were measured on-body inside an anechoic chamber using a Keysight PNA-X N5242A VNA. To minimize the computational resources, in the simulations, only the chest and arm parts of the body were examined. In measurements, the antenna performance was tested on a 3D flat phantom that had dielectric properties similar to human muscle tissues: relative permittivity (ε_r_) of 52.7 and 48.2, and conductivity (σ) of 1.95 and 6 S/m, at the lower and upper bands, respectively [29]. Figure 10 demonstrates the simulated and measured radiation patterns in various scenarios for the lower and upper bands, respectively. Usually, the body-arm is a suitable location for a wearable antenna because the radiation in the horizontal positions from the arm may not be easily affected, due to it being less surrounded by the lossy tissues than the chest position. It can be seen that the wearable antenna demonstrates omnidirectional radiation at the lower band, which can be used for on-body communication.

Similarly, omnidirectional radiation was also achieved at the upper band, which may be suited for off-body communications. For on-body S_11_ measurements, the subjects were chosen such that their body sizes were similar to the sizes of the chest and arm body models we used in the anechoic chamber. In practical situations, the performance of textile antennas may be affected by the fabrication tolerances and irregularities of the materials. It can be seen that the design was tested in different scenarios and maintained a wide BW with good stability in operating frequency and radiation patterns. Additionally, the measured gain, as reported in Table 3, showed stable values in both bands.

### 3.3. Specific Absorption Rate (SAR)

In WBAN, SAR is a key component for safety measurement, which can be measured according to the standard as mentioned by Usman Ali et al. in [30]. SAR shows how the electromagnetic waves are absorbed by the tissues that can cause a temperature rise.

In order to control the radiation damage to the human body, the SAR value should be lower than 1.6 W/kg according to the 1 g (Federal Communication Commission (FCC, U.S. standard) [31], or lower than 2 W/kg following the 10 g (ICNIRP, European standard) [32]. However, high SAR values are usually obtained if the antenna is located close to the skin surface. In measurements, the gap between the model and the antenna was kept at 10 mm to be similar to our assumption in the simulation, which reduced the SAR values as well as the loading effects of the body. Average SAR values for 1 g and 10 g tissue at both bands were calculated; they were lower than the standard limits for an input power of 20 dBm (0.1 W). Table 4 shows different SAR values calculated using CST MWS. Three scenarios were studied, i.e., antenna placed (1) in free-space, (2) on-chest, and (3) wrapped around the arm. The truncated ground with the ShieldIt layer was enough for protecting the body from unwanted radiation hazards. Maps of SAR values for these scenarios with the assumption of the maximum input power of 0.1 W in both bands are illustrated in Figure 11.

## 4. Free-Space and On-Body Link Budget

In body-centric communications, it is important to examine the transmission path loss (S_21_) of the antennas to ensure a reliable link budget. We measured S_21_ for several cases of free-space and on-/off-body communications, in an open space (parking area) at UTP. This measurement operation allowed us to show a range of possible values of the path loss. The path loss in terms of the transmission coefficient ($\left|S_{21}\right|$ dB) is the ratio of the input power (at transmitter antenna port) to the power received (at the receiver antenna port):(1)$$PL_{dB}\left(r\right)=-{\left|S_{21}\right|}_{dB}.$$

### 4.1. Measurement Setup and Procedure

The proposed dual-mode antenna’s link budget was measured in the free-space and on-/off body modes in an open space environment. We performed path loss measurements at different frequencies, considering the lower band of 2.2 GHz and the upper band of 5.6 GHz, using two meander line monopole antennas. The measurements were performed for the line-of-sight (LoS) and non-line-of-sight (NLoS) scenarios at different distances (r) and locations between the transmitting (Tx) and receiving (Rx) antennas. The measurement environment had a width of approximately 20 m. The receiving antenna was connected to an MS2720T spectrum analyzer that worked at zero spans. The Tx signal contained a single synthesized sweeper. The Tx was fixed, while the Rx was moved between 2 m to 11 m distances to analyze the path loss from measuring the transmission coefficient (S_21_) in the free-space and the on-body communication scenarios. It was ensured that all the surrounding objects were at a stationary state, and it was confirmed that the measurement was carried out accurately. The measurement setup including the cables was calibrated and a two-port calibration (without antennas) was performed to remove the ripple effects from the cables. Figure 12 shows the equipment setup for the path loss measurement.

### 4.2. Free-Space Communication

As a reference, measurements of path loss (S_21_) were initially carried out in free-space in a parking space, as illustrated in Figure 13. Tx and Rx antennas were installed on two tripods at the height of 1 m above the ground, with a proper vertical alignment. Besides, in S_21_ measurement, the distance (r) was set at 2 m, 5 m, 8 m, and 11 m to estimate the path loss in an open area. Due to large distances, it is not feasible to conduct a full-wave simulation; instead, we considered the open space parking lot measurement to estimate the path loss exponent. The log-distance path loss model was considered, which can be expressed as:(2)$$PL_{dB}\left(r\right)=PL\left(r_{0}\right)+10nlog\left(\frac{r}{r_{0}}\right),$$
where n is the path loss exponent, $r_{0}$ is the close-in reference distance, and r is the separation between Tx and Rx [33]. Using $r_{0}=2\;m$ and r=11 m, we obtained an average n=2.95, which is well-matched with the urban areas [33,34].

### 4.3. On- and Off-Body Communications

In the on- and off-body communication cases, measurements at the lower and upper bands were performed. For this part, the antennas were placed on various locations of the body (chest and arm) to measure the S_21_ values, as shown in Figure 14. To investigate the effect of the distance (r) on the S_21_ parameter, the antenna separation distances were set at 2 m, 5 m, and 8 m between the Tx and Rx antennas with a gap of 10 mm created between the antenna and the skin layer to mimic the real scenario, the same as mentioned in the S_11_ measurement.

In the first set of measurements, two identical antennas were placed on the chest of two male subjects, with an average weight of 72 kg and height of 1.58 m. The arms were stretched along the body of the subject. The path losses in the presence of the body are larger due to the power absorption of lossy human body tissues, and the S_21_ levels were reduced in comparison to the free-space measurements. Similar to the free-space measurements, the distance between the Tx and Rx antennas was increased from 2 to 8 m for on- and off-body communications. Using Equation (1) and $r_{0}=2\;m$, and r=8 m, we estimated the value of n=3, which was slightly larger than the free-space case. This was expected due to the obstruction by the bodies and the losses and power absorbed by the lossy tissues. We also examined one NLoS case, where the antennas were placed on the chest and the arm of two male subjects with a separation of r = 2 m. We observed more path losses of 90 dB and 95 dB in the lower and upper bands, respectively. In the next scenario, the antennas were attached to the arms of the two male subjects, and LoS path losses were measured. Path losses of about 80 dB and 86 dB were obtained in the two bands, respectively.

Overall, the path losses in the three scenarios of free-space, on-chest, and on-arm at lower bands were lower than those for the off-body at the upper band. To comply with the SAR regulations [35], the antenna input power was kept at 20 dBm. For example, when the receiver showed a received power of −75 dBm [36,37], the maximum loss of 99.4 dB (S_21_ = −99.4 dB) was considered to be acceptable. The measurement results showed that the lower band has better performance and can be used for LoS, and the upper band might be better suited for the NLoS. Thus, our proposed antenna provides suitable path losses in an open-space in both bands and can be used for the link budget calculations. An average path loss exponent of n = 3 was observed.

## 5. Conclusions and Future Directions

In this study, a compact wearable flexible dual-band planar meander line monopole was designed for a wide bandwidth (2.2–3.0 GHz and 5.6–6.0 GHz). The proposed antenna was evaluated to validate its performance in various scenarios such as in free-space and on the body with different sizes and under different bending radii and wet conditions. The proposed design was fabricated, and the measurement results agreed well with the simulation results. The proposed antenna provided wide bandwidth and omnidirectional radiation patterns with an acceptable gain values in both free-space and on-body cases. The variation in the impedance matching and operating frequency bands were bounded by 0.5 and 100 MHz, respectively, in both bands, while placed in the vicinity of the human body at an optimum distance of 10 mm from the skin layer. Moreover, the antenna was examined, so that the maximum current of the monopole strip was concentrated on the meander line. In the future, the resilience of the proposed design will be further studied for longer distances in the lossy environment using highly flexible, stretchable 2D high-conductive transition materials such as MXene ($Ti_{3}C_{2}T_{X}$) [38], and niobium diselenide ($NbSe_{2}$) [39], as well as flexible substrate materials such as embedded Polydimethylsiloxane PDMS [40].

## Figures and Tables

**Figure 1 micromachines-12-00475-f001:**
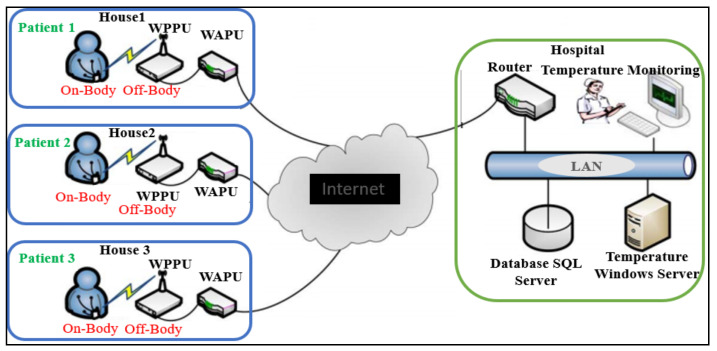
A general healthcare architecture for integrated meander line antenna. WPPU: wireless patient portable unit; WAPU: wireless-access point unit; LAN: local area network; QoS: quality of service [7].

**Figure 2 micromachines-12-00475-f002:**
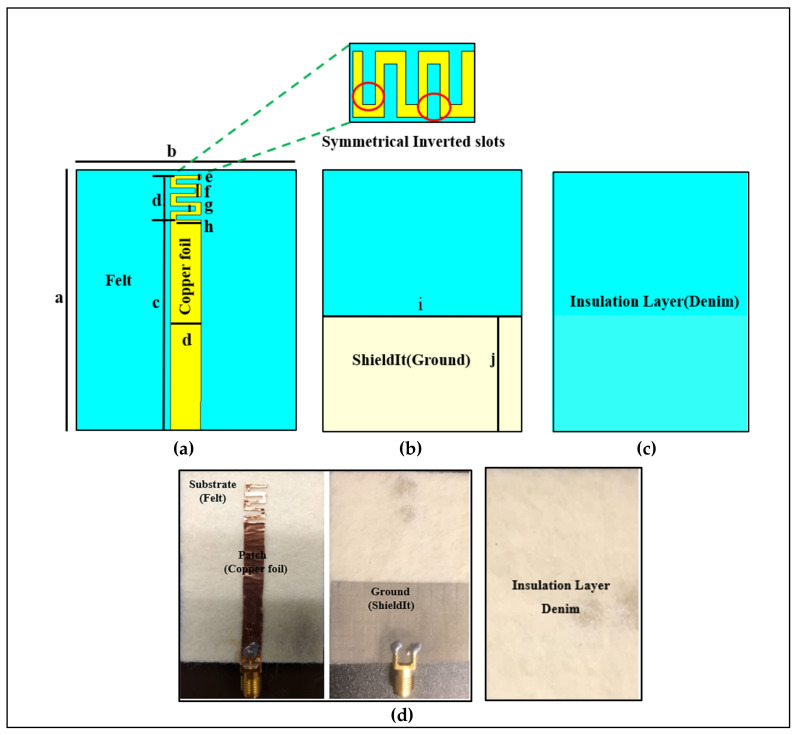
Design and dimensions of the planar meander line antenna, (dimensions are given in mm): (**a**) the top side; (**b**) the bottom side; (**c**) the insulation layer; and (**d**) the fabricated antenna.

**Figure 3 micromachines-12-00475-f003:**
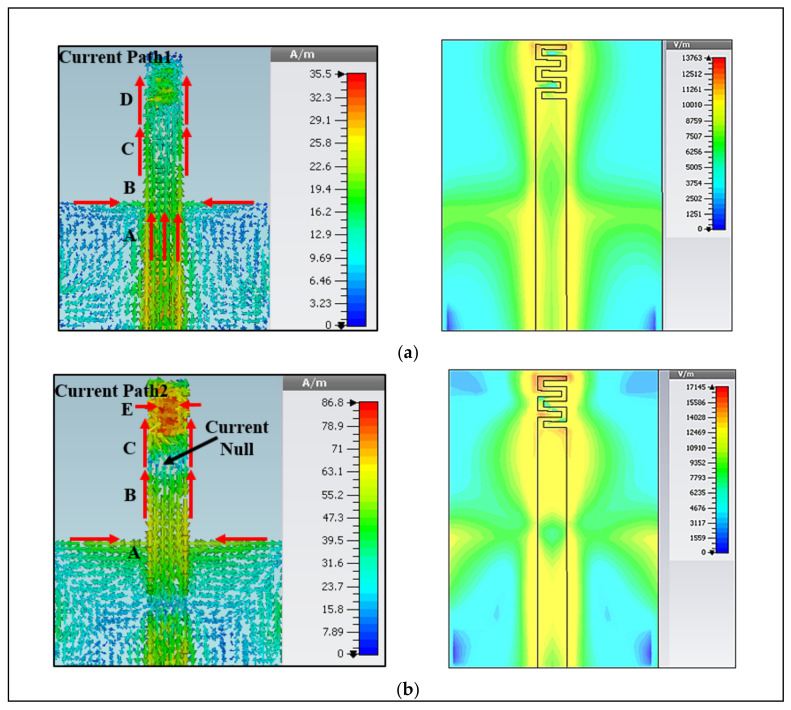
Distribution of current density with radiation modes at (**a**) lower band, and (**b**) upper band.

**Figure 4 micromachines-12-00475-f004:**
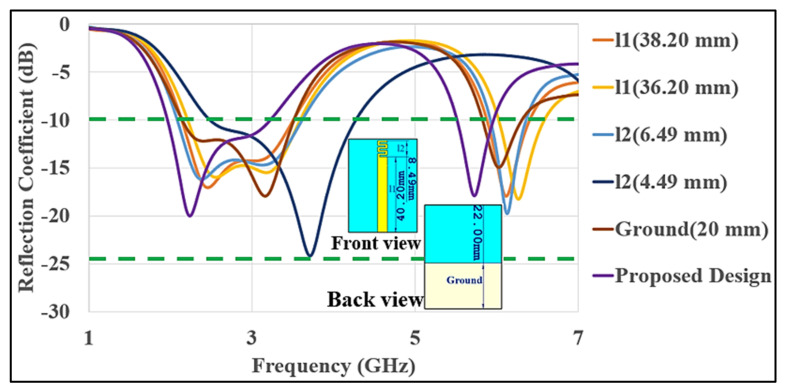
Variation in the S_11_ by changing the length ($l_{1}{\;\mathrm{and}\;l}_{2}$) of the proposed antenna design.

**Figure 5 micromachines-12-00475-f005:**
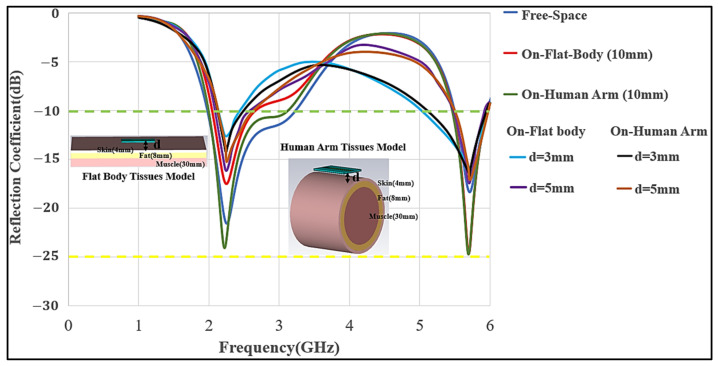
Optimization of S_11_ for meander line monopole antennas (MMAs) in free-space, on-chest flat model, and arm model with various locations on the body (3 mm, and 5 mm).

**Figure 6 micromachines-12-00475-f006:**
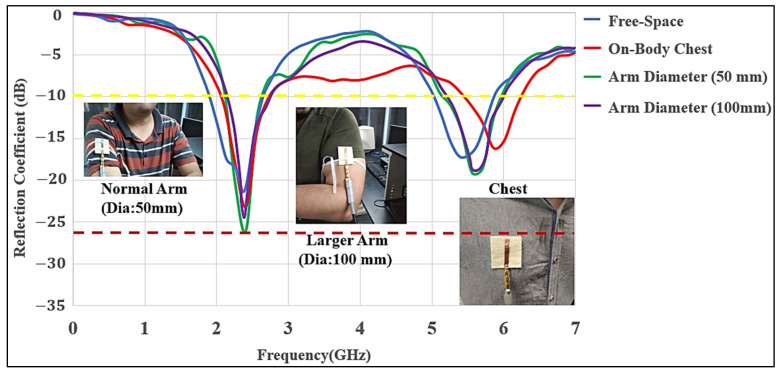
Measured S_11_ of the MMAs in free-space and on different locations of the human body (chest, 50 mm; small size arm, 50 mm diameter; and large size arm, 100 mm diameter).

**Figure 7 micromachines-12-00475-f007:**
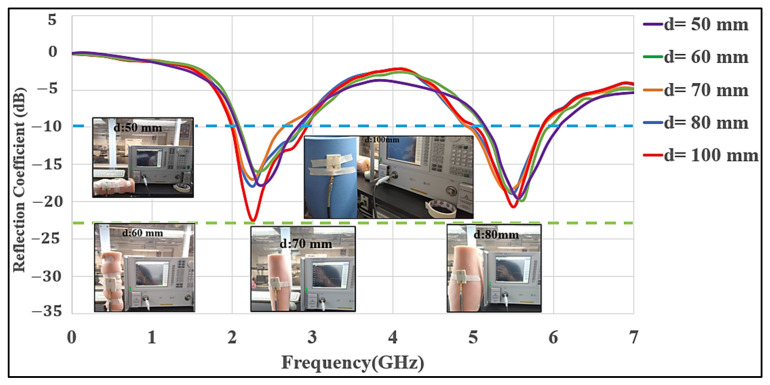
Measured S_11_ of the MMAs deformed on various cylindrical curvatures (d = 50 mm, 60 mm, 70 mm, 80 mm, and 100 mm).

**Figure 8 micromachines-12-00475-f008:**
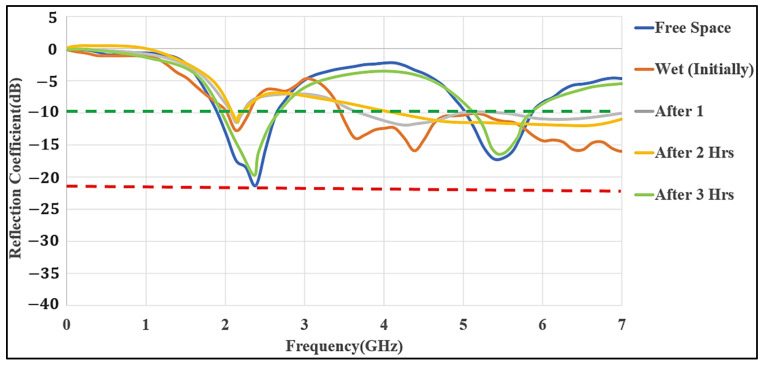
Measured S_11_ of the MMAs in wet conditions (Ref (free-space), initial condition, after 1 h, 2 h, and 3 h).

**Figure 9 micromachines-12-00475-f009:**
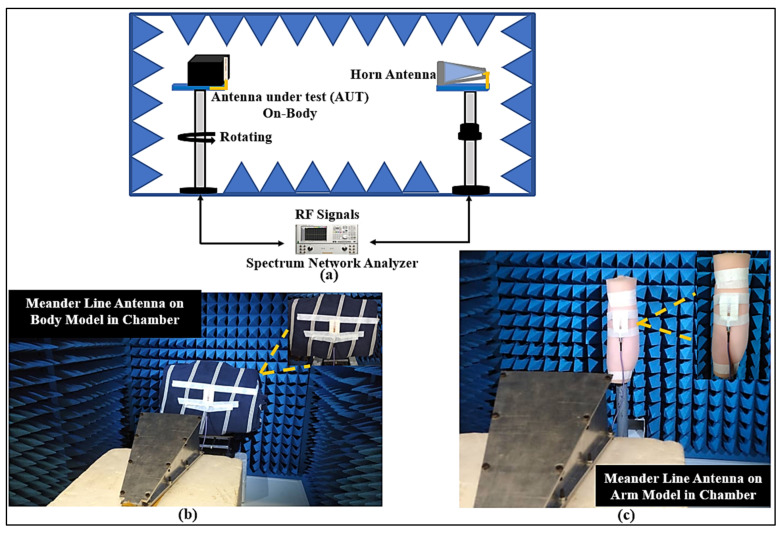
(**a**) Setup for the 3D far-field measurement for meander line antenna in the anechoic chamber, (**b**) human chest model, and (**c**) human arm model.

**Figure 10 micromachines-12-00475-f010:**
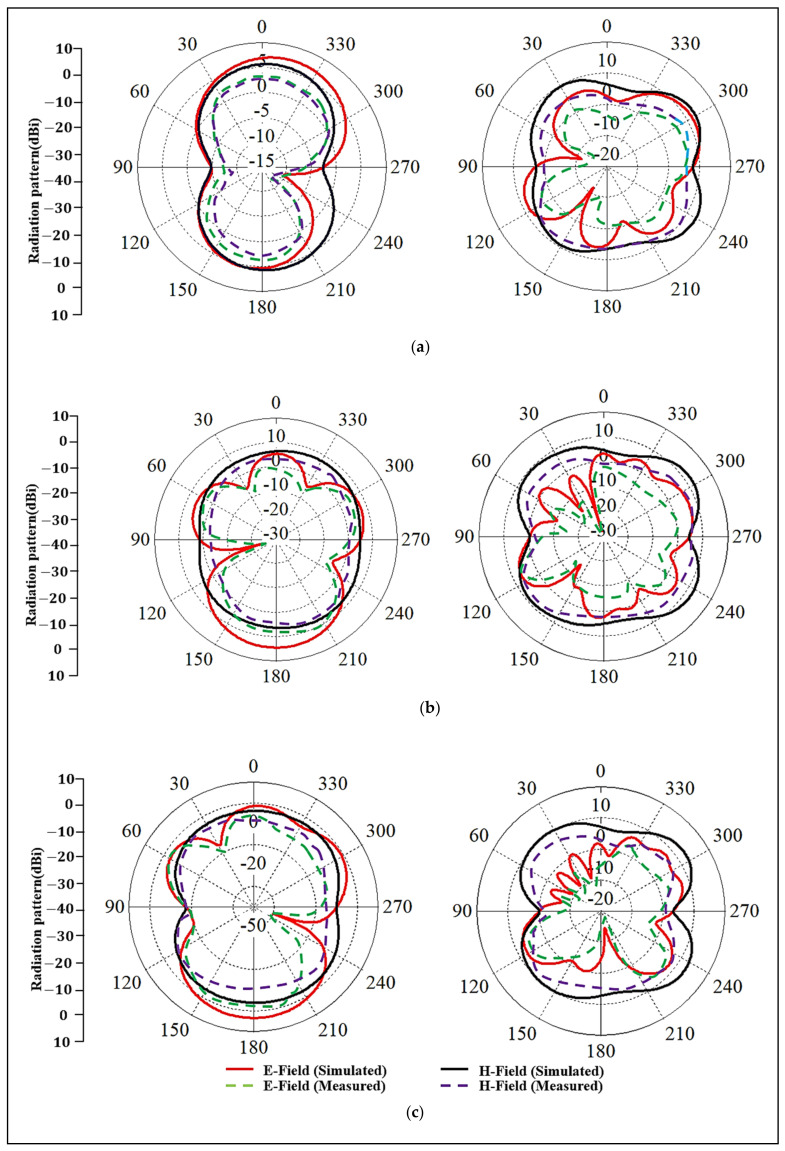
Simulated and measured radiation patterns of E and H planes at both bands, left side: the lower frequency, right: the higher frequency, (**a**) free-space (2.2 GHz and 5.6 GHz), (**b**) on-chest (2.3 GHz and 5.7 GHz), (**c**) on-arm (2.2 GHz and 5.7 GHz).

**Figure 11 micromachines-12-00475-f011:**
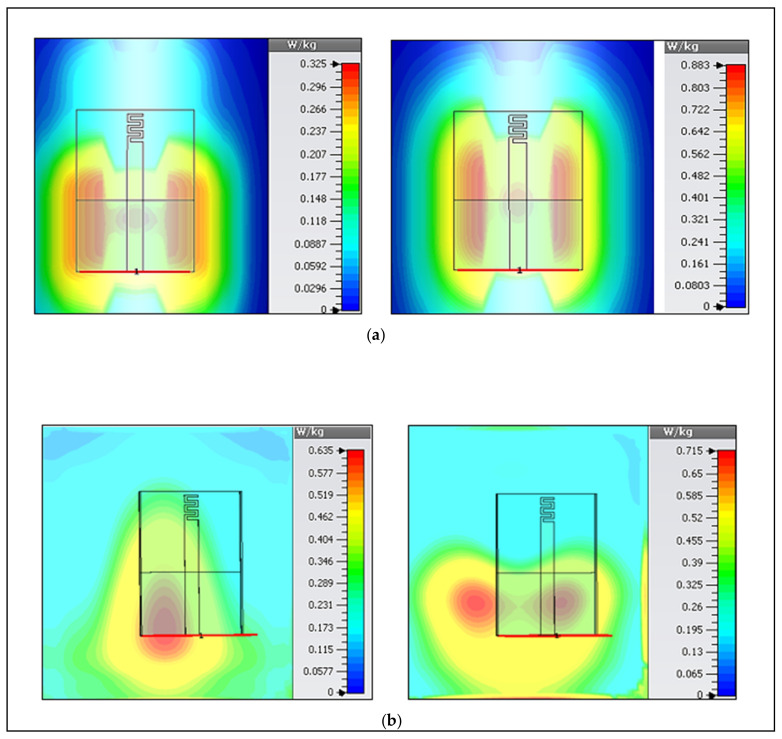
Simulated SAR (10-g average) results when the antenna is on the (**a**) upper-right arm of a male subject (at 2.2 GHz (left) and 5.7 GHz (right)), and (**b**) on the chest center (2.3 GHz (left) and 5.7 GHz (right)).

**Figure 12 micromachines-12-00475-f012:**
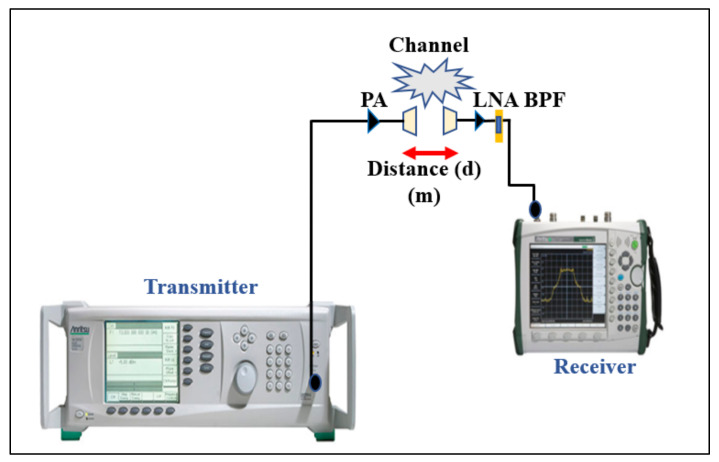
Measurement equipment setup for the path loss (LNA: low noise amplifier; BPF: band pass filter; PA: power amplifier).

**Figure 13 micromachines-12-00475-f013:**
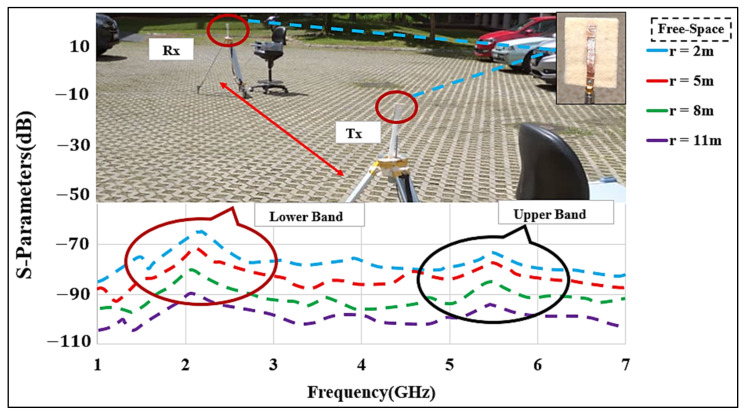
Measured S_21_ of the MMAs in free-space at various distances between the Tx and Rx antennas (r: 2 to 11 m).

**Figure 14 micromachines-12-00475-f014:**
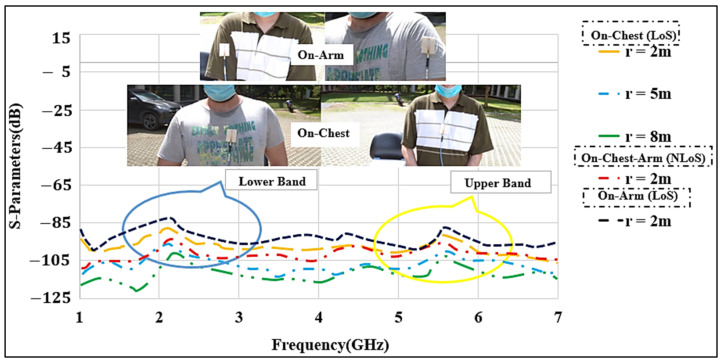
Measured S_21_ of the MMAs for on-chest and on-arm in line-of-sight (LoS) and non-line-of-sight (NLoS) at various distances between the Tx and Rx antennas (r: 2 to 8 m).

**Table 1 micromachines-12-00475-t001:** Comparison of operating bands, materials, size, bandwidth (MHz), bending effect, and specific absorption rate (SAR) against other flexible antennas published in the literature.

Ref.	Frequency (MHz/GHz)	Material Substrate/Patch	Size (W × L) (mm^2^)	Bandwidth (BW) (MHz)	SAR (W/Kg) (1 g/10 g)	Description
[14]	2.4 Sleeve-Badge Textile Antenna	Fully Textile (Multiple Layers)	50 × 57	140.0 (Free Space)	0.34 (10 g)	Fully-textile, large size, single band. Not fully tested on the body.
[15]	2.5/4.2/5.5/6.8 Defected Ground Structure (DGS) with Meander line	Copper/FR4	40 × 40	70/90/350/710 (Free Space)	NA	Multiband, small size, not fully textile. Might be difficult to fabricate.
[16]	2.4/5.2 Metamaterial-loaded Antenna	Felt/ShieldIt	50 × 50	85/200 (Free Space) 130/698 (On-Chest)	0.2/0.15 (On-Arm) 0.12/0.25 (On-Chest)	Fully textile, dual-band, small size, measured on body, use of metamaterial might make it difficult to fabricate.
[17]	2.5 Monopole Patch antenna	ShieldIt/(Jeans Cotton)	50 × 60	667 (Free-Space)	NA	Fully textile, small size, single band, Not tested on the body.
[18]	2.4 U-slot Patch Antennas	Felt/Copper Foil	85×70	270 (Free space)	NA	Fully flexible, single band, large size, mostly evaluated performance by substrate thickness.
[19]	2.075 to 2.625 Monopole Antenna with FSS Surface	Jeans/Copper Foil	38 × 50.8	550 (Free-Space)	3.03 (10 g)	Fully flexible, small size, use of frequency selective surface might make it difficult to fabricate, single band, not fully tested on the body and under bending.
[20]	2.4/5.5 Meander Line Slots Antenna	Felt/ShieldIt	82 × 72	132/422 (Free Space) 138/409 (On-Body)	0.16/0.2 10 g (On-Chest)	Fully textile, dual bands, large size, not fully tested on the body.
[21]	2.4 GHz Microstrip Patch Antennas	Felt/ShieldIt Super	80 × 100	101 Free Space 103/103 (Right-Arm/Chest)	2.88/0.35 (Upper Right Arm)	Fully flexible, single-band, not fully tested on the body. Used for exposure to the physiological parameters.
[22]	2.45 L-Shaped Patch Antenna	Polyphenylene Ether (PPE)/Copper	50 × 50	120 (On-Body)	0.6414/1.524 10 g (On-Chest)	Large size, single band. Not flexible, not fully tested on the body.
[23]	2.45 Full Flexible Monopole Antenna	Felt/ShieldIt	40 × 60	720 (Free space) 550 (On-arm) 620 (On chest)	NA	Fully flexible, single-band, large size, not fully tested on the body.
**This work**	2.20–3.00/5.60–6.00 Meaner Line Monopole Antenna	Substrate (Felt)/Copper Tape/ShieldIt (Patch and Ground)	37.20 × 50.0	1282.4/450.5 (Free space) 688.9/500.9 (On-Chest) 1261.7/524.2 (On-Arm)	On-Chest 1.22/0.75 (1 g), 0.63/0.71 (10 g) On-Arm 1.38/0.692 (1 g), 0.883/0.325 (10 g)	Fully flexible, compact size, dual-band, tested in free-space and on the body with various bending and wet conditions. High stability and high degree of isolation in two bands.

**Table 2 micromachines-12-00475-t002:** The dimensions of the proposed meander line monopole antenna.

Symbols	Dimensions (mm)
a	50.00
b	37.20
c	40.20
d	8.49
e	0.73
f	2.38
g	1.00
h	14.13
i	37.20
j	22.00

**Table 3 micromachines-12-00475-t003:** Summary of the antenna’s performance measures.

Characteristics	Dual-Band Monopole Meander-Line Antenna (Lower/Upper Band)		
Antenna Parameters	Free-Space	On-Chest	On-Arm
Simulated S_11_ (dB) at the Center Frequency	−21.02/−17.00	−17.73/−17.00	−25.20/−25.00
Measured S_11_ (dB) at the Center Frequency	−21.00/−19.00	−23.00/−19.00	−25.00/−20.00
Simulated and Measured Center Frequency (GHz)	2.2 and 5.6/3.0 and 5.6-6.0	2.3 and 5.7/3.0 and 6.0	2.2 and 5.7/3.0 and 6.0
Simulated Bandwidth (MHz)	1282.4 and 450.5	688.9 and 500.9	1261.7 and 524.2
Measured Bandwidth (MHz)	1255.0 and 430.2	671.9 and 475.2	1243.1 and 501.1
Simulated Gain (dBi)	3.03/4.85	3.80/4.67	3.00/4.55
Measured Gain (dBi)	2.76/3.67	3.45/4.25	2.76/4.15
Simulated Radiation Efficiency (%)	Lower (85.34/83.32) Upper (63.82/50.57)	(54.37/53.40) (54.27/51.39)	(60.0/59.51) (63.73/62.05)
Voltage Standing Wave Ratio (VSWR)	1.22/1.29	1.61/1.21	1.06/1.19
Radiation Pattern (O/O)	Omnidirectional	Omnidirectional	Omnidirectional

**Table 4 micromachines-12-00475-t004:** Comparison of different SAR values.

**Frequency** **(GHz)**	**1-g**	**10-g**	**1-g**	**10-g**	**SAR Limits** **(1-g US/10-g EUR)** **(1.6/2 (W/Kg))**
	**(On-Body)**		**(On-Arm)**		
2.2–3.0	1.22	0.63	1.38	0.883	
5.7–6.0	0.75	0.71	0.692	0.325	

Input power to the monopole meander line antenna was 0.1 W or 20 dBm.

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
