# Peer review of "Design and Evaluation of a Flexible Dual-Band Meander Line Monopole Antenna for On- and Off-Body Healthcare Applications"

_micromachines, 2021, doi:10.3390/mi12050475_

Round 1
Reviewer 1 Report
The authors present a WBAN antenna operating in the frequency bands 2.2-3 GHz and 5.6-6 GHz possessing minimized dimensions and omnidirectional radiation pattern for both the operating bands. Various and exhaustive numerical and experimental analysis have been carried out on the presented antenna, showing results agreement between simulations and measurements and good operating performances.
The paper is clear and well written (apart from some minor adjustments needed on the English).
Some questions:
- I strongly suggest to add further information and explanation in the captions of the figures that show measurements or S-parameters comparisons. For example, I suggest to report the parameters (with their names) also in the figures caption, in order to make them more easily readable and understandable.
- In subsection 2.4 the authors say that they choose to simulate the antenna at a distance of 10 mm from the human body models, in order to not impact on the antenna impedance matching. In section 3, they state that this distance is necessary for safety. However, in section 3.1 and in particular from figures 6 and 7, it is unclear if this distance is maintained in the shown measurements. Please better explain this point.
- Again, for the sake of clarity, it could be useful to extract the inset photos of figures 13 and 14 and report them as a stand-alone figure with higher resolution.
Author Response
We thank the reviewers for the time spent on the review and the comments. Thanks once again.

Reviewer 2 Report
This paper claims to present a compact wearable flexible meander line monopole antenna for on and off body healthcare applications. My comments on this work are as follows:
(1) Introduction is too long with irrelevant kinds of stuff. It should be short and have a recent state of the arts only.
(2) The abstract did not provide any contribution to the manuscript. It should contain kinds of stuff that is original.
(3) the author claims to have the antenna with a compact structure but there are a lot of antennas much more compact than this.
(4) The structure and design of the antenna are not novel. This sort of structure is published in literature again and again.
(5) after reading the whole article, it looks like a book chapter that discusses antenna parameters and characteristics that every antenna researcher knows. There should be new kinds of stuff or some contribution to this work. Mere calculating antenna different characteristics did not qualify a paper as an original article.
(6) Figure 2: dimensions are not professional and important other dimensions are missing.
(7) Fig. 3: color bars value of surface radiation patterns is unclear.
(8) Figure 4 is redundant and did not give any information. It is well known that how length affects the frequency bands. It will be much more professional if it's provided in terms of s-parameters.
(9) Comparison table: Recent state of the art that has better results than the current design. authors have ignored them. you can search literature thoroughly;y for this.
(10) Figure 7 is somewhat suspicious and wrong. Provide snapshots from VNA.
(11) Radiation patterns are not good. there are too many ripples even the frequency is very low.
(12) Conclusion should be short and must reflect abstract. You can do your discussion in the above sections as well.
Author Response

(The authors gave the same response as above.)

Reviewer 3 Report
In general, this paper is well written and consists of a lot of technical details. Please consider my comments below
- As a technical paper, this reviewer thinks that the introduction section should be condensed and make it shorter with less number of references. Overall the novelty is not very much in terms of the antenna geometry itself. It is then of great importance to highlight the contribution of this work to attract the readers' attention. Instead of reporting or summarizing the performance of others' work, the authors should explain the key reason that achieved such minor change in the antenna performance for the cases tested in free space and on body.
- Discussion of SAR has been pretty much emphasized. It is not clear why this figure of merit plays such an important role. Please comment on this.
Author Response
We thank the reviewer 3 for the time spent on the review and the comments.

Reviewer 4 Report
The presented study starts with a comprehensive analysis of the state-of-the-art. Although many groups have been working on the important and emerging field of wearable flexible antennas, the aim and the benefit of the authors's research is clearly shown. The design method is clearly described. Also the electromagnetic behavior is well explained. Comprehensive experiments with big practical relevence have been conducted. The results are convincing and well described. Thus, the paper will be a benefit to the community.
Some aspects of the visual presentation and language could be improved:
- Fig. 5: unit is missing for d in the Figure
- Fig. 10: the font size for the legend is very small
- Fig. 11: the font size of the color bar numbers is very small
- some typos, e.g. line 207 "fields distributions", line 330 "perfromance"
Author Response
We thank the reviewer 4 for the time spent on the review and the comments.

Round 2
Reviewer 2 Report
Thanks for revising the manuscript. Some of my major concerns are still not addressed properly and authors need to be careful while addressing the reviewer comments. My comments that I am still not satisfied with is mentioned in further details for the author's convenience:
(2) abstract is still very lengthy and contains irrelevant information. The contribution of the work is questionable as before as o new contributions were added in the revised version.
(3) the author claims to have the antenna with a compact structure but there are a lot of antennas much more compact than this. I still feel the same and it has nothing to do with a wideband as this is not a wideband antenna. Easy integration is not something that the author has shown. The majority of monopole antennas are easily integrable. There are antenna that did not suffer from bandwidth reduction, and if it is the case that antenna must not be considered at all.
(4) The structure and design of the antenna are not novel. This sort of structure is published in literature again and again. I feel the same after reading the authors' response as the same antenna structure is present in literature.
(5) after reading the whole article, it looks like a book chapter that discusses antenna parameters and characteristics that every antenna researcher knows. There should be new kinds of stuff or some contribution to this work. Mere calculating antenna different characteristics did not qualify a paper as an original article.
I believe that there is no need for such irrelevant stuff and the majority of antenna researchers know the basic issues. Authors are trying to extend the paper just for the sake to show their contribution. Mere increasing the length and introduction does not mean that manuscript is providing some information to the society and community. I highly disagree that "this paper is among one of the few papers that collectively considers and examines most of the important parameters for wearable flexible antennas in a practical context and as a prelude to characterizing the antenna various parameters in a practical operational environment for a wearable application in the vicinity of the human body."
Every good antenna paper considers these kinds of stuff. authors need to be careful and to the point while addressing important comments.
All other comments are acceptable and this reviewer thanks the authors for introducing it.
Author Response
We thank the reviewer 2 for the time spent on the review and the comments.

Reviewer 3 Report
This reviewer does not have any further comments.